# Maresin 1, a Proresolving Lipid Mediator, Ameliorates Liver Ischemia-Reperfusion Injury and Stimulates Hepatocyte Proliferation in Sprague-Dawley Rats

**DOI:** 10.3390/ijms21020540

**Published:** 2020-01-15

**Authors:** Gonzalo Soto, María José Rodríguez, Roberto Fuentealba, Adriana V. Treuer, Iván Castillo, Daniel R. González, Jessica Zúñiga-Hernández

**Affiliations:** 1Escuela de Tecnología Medica, Facultad de Ciencias de la Salud, Universidad de Talca, Talca 3460000, Chile; gonzalosooj17@gmail.com; 2Programa de Doctorado en Ciencias mención Investigación y Desarrollo de Productos Bioactivos, Instituto de Química de los Recursos Naturales, Universidad de Talca, Talca 3460000, Chile; mjrodriguezbecerra@gmail.com (M.J.R.); rfuentealbaleyton@gmail.com (R.F.); 3Escuela de Medicina, Universidad de Talca, Talca 3460000, Chile; 4Departamento de Ciencias Básicas Biomédicas, Facultad de Ciencias de la Salud, Universidad de Talca, Talca 3460000, Chile; atreuer@utalca.cl (A.V.T.); dagonzalez@utalca.cl (D.R.G.); 5Centro de Bioinformática, Simulación y Modelado, Facultad de Ingeniería, Universidad de Talca, Talca 3460000, Chile; 6Unidad de Anatomía Patológica, Hospital Regional de Talca, Talca 3460001, Chile; icastillo@ucm.cl; 7Centro Oncológico, Facultad de Medicina, Universidad Católica del Maule, Talca 3466706, Chile

**Keywords:** liver preconditioning, bioactive lipids, mitotic activity index, cytokines, transcription factors

## Abstract

Maresin-1 (MaR1) is a specialized pro-resolving mediator, derived from omega-3 fatty acids, whose functions are to decrease the pro-inflammatory and oxidative mediators, and also to stimulate cell division. We investigated the hepatoprotective actions of MaR1 in a rat model of liver ischemia-reperfusion (IR) injury. MaR1 (4 ng/gr body weight) was administered prior to ischemia (1 h) and reperfusion (3 h), and controls received isovolumetric vehicle solution. To analyze liver function, transaminases levels and tissue architecture were assayed, and serum cytokines TNF-α, IL-6, and IL-10, mitotic activity index, and differential levels of NF-κB and Nrf-2 transcription factors, were analyzed. Transaminase, TNF-α levels, and cytoarchitecture were normalized with the administration of MaR1 and associated with changes in NF-κB. IL-6, mitotic activity index, and nuclear translocation of Nrf-2 increased in the MaR1-IR group, which would be associated with hepatoprotection and cell proliferation. Taken together, these results suggest that MaR1 alleviated IR liver injury, facilitated by the activation of hepatocyte cell division, increased IL-6 cytokine levels, and the nuclear localization of Nrf-2, with a decrease of NF-κB activity. All of them were related to an improvement of liver injury parameters. These results open the possibility of MaR1 as a potential therapeutic tool in IR and other hepatic pathologies.

## 1. Introduction

Liver ischemia-reperfusion (IR) injury is a consequence of the restoration of flow after an occlusion of the vasculature. The subsequent injury is directly related to the duration of the ischemia and is an important cause of morbid-mortality due to liver transplant and resection [1,2]. Hepatic IR injury remains a fundamental and unavoidable component of liver transplantation. The postoperative graft dysfunction after liver transplantation is associated with cold ischemia time (cold graft preservation) and warm ischemia time, defined as a rewarming time. Finally, the organ recovery process is followed by reperfusion, resulting in the IR phenomena [3].

During the early phase of IR, the generation of reactive oxygen species (ROS) causes damage to hepatocytes through lipid peroxidation, protein oxidation, mitochondrial dysfunction, and DNA damage. This is triggered by oxygen deprivation followed by a restoration of oxygen delivery, which implies endothelial cells’ and hepatocytes’ death [4]. Additional factors that contribute to the damage are Küpffer cells (KCs), neutrophil infiltration and lymphocytic activation, increasing ROS, protease liberation, and cytokines’ increment [5]. Despite the contribution of oxidizing agents derived from KCs, the extent of injury during this initial phase is much lower than that observed in later stages. Events that occur during the initial phase of liver damage, including the activation of KCs, initiate a complex inflammatory cascade leading to the recruitment of several leukocyte populations to the liver throughout reperfusion [1].

Polyunsaturated fatty acids (PUFA) are important dietary components with key roles in several physiological functions. Omega-3 fatty acids, such as α-linoleic, eicosapentaenoic acid (EPA), and docosahexaenoic acid (DHA), present important properties for which they can be classified as functional foods [6]. Both EPA and DHA have been reported as effective anti-inflammatory and tissue-protective mediators. In addition, the metabolism of eicosanoids plays a crucial role in the anti-inflammatory pathways related to the conversion of EPA and DHA to E-resolvins and D-resolvins (series E and D, respectively) [5,6]. Resolvins have potent immunomodulatory properties, limiting inflammation through several mechanisms, including reduction of proinflammatory eicosanoids synthesis, decrease in leukocyte chemotaxis, inhibition of expression of adhesion molecules, and reduced activation of NF-κB [7,8].

Maresin-1 (MaR1) is a recently discovered DHA derivative [7]. In vitro and in vivo evidence indicate that MaR1 promotes the resolution of inflammation and exerts cytoprotective effects via inhibition of neutrophil infiltration and selective stimulation of macrophage phagocytosis. Also, MaR1 partially inhibits the generation of ROS through mitogen-activated protein (MAP) kinase signaling [8] and reduces the inflammatory response. In the liver, the MAP kinase pathway can be activated by cytokines and ROS [9]. This activation is critical for the maximal expression of some inflammatory cytokines (e.g., interleukin (IL)-1β, monocyte chemoattractive protein (MCP)-1, and tumor necrosis factor (TNF)-α), which play important roles in liver damage [8]. Considering the protective effects of MaR1, the main objective of this study is to determine the hepatoprotective action of MaR1 in liver injury and oxidative stress in a rat model of IR. 

## 2. Results

To test for the potential of MaR1 to attenuate liver damage, a group of rats was submitted to hepatic IR. As expected, a significant increase in serum AST and ALT transaminases was observed after IR (Figure 1A,B). The effect of MaR1 administration was tested for its capacity to decrease AST and ALT at 3 h post-reperfusion. As observed, IR increased the levels of these transaminases 2–5-fold compared to the control-sham and MaR1-sham groups (*p* < 0.05). Interestingly, the MaR1-IR group showed a slight but significant decrease in the level of both transaminases (*p* < 0.05) related to the IR group. 

The histological assessment of the livers from the sham and MaR1-sham groups showed preserved liver morphology, portal space with low inflammatory lymphocytic infiltrate, venules, bile duct, and central veins are identified and they do not have alterations (Figure 2). On the contrary, the IR group presented loss of integrity and architectural distortion, with some diffuse inflammatory cells, mostly related to polymorphonuclear (PMN) and neutrophil infiltration with centrilobular necrosis focus, related the damage produced by the IR process. On the contrary, Mar1-IR livers exhibit a normal cytoarchitecture, with a significant decrease of the inflammatory foci, with minimal-to-moderate necrosis compared to the IR group. The IR damage was quantified as cytoarchitecture, inflammation, and necrosis (Figure 2E–G). The IR group showed altered values for the architecture, inflammation and necrosis parameters when compared to controls. The MaR1-IR group normalized this morphological observed damage induced by IR. Next, we evaluated the mitotic index (MAI) (Figure 3), which showed that the mitotic cell count was increased in both groups that received MaR1. MAI activity of hepatocytes was detected 3 h post-reperfusion and was characterized by an intense cell division with 3.7- and 5.25-fold increases in the MaR1-sham and MaR1-IR groups, respectively. Also, MaR1-IR showed an increase of 41% in cell division related to MaR1-sham livers. The MaR1 impact on liver regeneration was evaluated using Ki67 immunofluorescence (Figure 4), an indicator of cell proliferation, observing an augmented signal in the groups treated with MaR1 compared to sham (*p* < 0.05), but IR shows a 2-fold decrease in the Ki67 signal in comparison to the sham group.

We evaluated the impact of MaR1 in the process of necroptosis (Figure 5) using the expression of receptor-interacting protein 3 (RIP3). This marker did not show statistical variations among the groups. In addition, protein expression of cleaved caspase-3, a marker of active apoptosis, increased 1.8-fold in the MaR1-IR group compared with the others, while the other groups did not show statistical differences among them.

Next, the levels of the inflammatory cytokines were evaluated (Figure 6A,B). TNF-α and IL-6 showed an 8.2- and 9-fold increase after IR over control rats. MaR1 treatment reduced TNF-α levels by 3.1 compared to IR groups (*p* < 0.05), but still exceeded the control values by 2.3 times. IL-6 was increased 1.4 times in the MaR1-IR group compared to IR groups (*p* < 0.05). Serum IL-6 was also elevated 2.1 times in MaR1-sham with respect to the control and was 0.2 and 6 times less than the IR and MaR1-IR groups. Levels of the anti-inflammatory IL-10 (Figure 6C) were decreased 0.8 times in IR compared to sham, and MaR1 increased these levels by 1.6 and 3.5 for MaR1-sham and MaR1-IR, this last group shows an increase of 4.2 times compared to IR alone.

The pan-marker of macrophages, F4/80, was assayed by immunofluorescence (Figure 6D,E) as an indicator of hepatic immune cells present. IR liver showed an increase of F4/80 signal of 1.9-folds in comparison to the sham group, and MaR1 administration increased 4.0-fold in relation to IR (*p* < 0.0001) and 7.5-fold and 15-fold in relation to control Sham and MaR1-sham, respectively. Interestingly, MaR1-sham showed a decrease in the signal of F4/80 compared to control sham.

The expression of the transcription factors Nrf-2 and NF-κBp65 and their nuclear translocation was investigated (Figure 7), since these factors are targets of the inflammatory cytokines. At three hours of reperfusion, NF-κB p65 nuclear translocation in liver subjected to IR was increased by 100% (*p* < 0.05) in relation to controls (sham-operated animals). This effect was suppressed by the administration of MaR1, without significant changes compared to control groups (Figure 7B,D). The observed increase of NF-κB-p65 is associated with a decrease in the total of Nf-κB p65 found in the cytoplasm in comparison to the control (*p* < 0.05) (Figure 7A,C). Then, we analyzed the nuclear translocation of Nrf-2, the IR group shows cytoplasm levels of Nrf-2 similar to those observed in MaR1-sham groups, and a small amount of Nrf-2 migrated to the nucleus (Figure 7B). Nuclear Nrf-2 was incremented only in the MaR1-IR group (*p* < 0.05) and the increase was more than 7-fold with respect to the control (Figure 7B).

## 3. Discussion

In the present study, we show for the first time the ability of MaR1 to ameliorate the deleterious histopathological changes generated by IR related to pro-inflammatory cytokine signaling. Also, it was found that MaR1 can induce hepatic cellular division in IR and modulate NF-κB and Nrf-2 transcription factors.

Liver surgery and transplantation are the most effective treatment of choice for patients with end-stage liver disease and for benign and malignant tumors. One of the complications of surgery is restoring the blood supply after ischemia, where the liver is prone to further injury that aggravates the damage already caused by ischemia [10]. There are several strategies under development related to improve the IR damage, including improvements in donor organ perfusion and preservation methods, pharmacologic treatments, gene therapy, and surgical techniques, among others. There is evidence that IR not only generates complications, but also ameliorates survival in an animal model [11]. For the aforementioned reasons, IR surgery is a still a challenge in the clinical setting.

MaR1 increased the mitotic and Ki-67 labeling index, which suggests that this treatment promoted the cell cycle and cell proliferation. It is important to consider that MaR1 is a derivative from DHA, an omega-3 fatty acid [12,13,14]. Omega-3 fatty acids can play a protective role in liver function after partial hepatectomy (PH), with improvement in liver regeneration and functional recovery following PH in experimental models [15]. Also, omega-3 fatty acids are responsible for β phosphorylation of activation of protein kinase B (Akt). Signal transducer is important in the signaling pathway involved in regulation of cell cycle, proliferation, transformation, and apoptosis of hepatocytes [16]. Along with the aforementioned, among the specific pro-resolving molecules, the members of the maresin family are considered as important molecules for tissue regulation, with potent pro-resolving, tissue-regenerative, and immunoresolvent activity in human organs and tissues, including lymph nodes and serum [17]. IR is a complex phenomenon involving massive inflammation and cell death, including apoptosis and necrosis with the consequent organ dysfunction. 

Necroptosis is a novel mode of cell death, known as “programed necrosis”, which is controlled by two kinase receptor-interacting proteins (RIP1 and RIP3) [11]. RIP3 and the necrosome have shown controversial results in IR. On one hand, it has been found that necroptosis could be setting in TNF-α-induced cell death, and a recent study has found that necroptosis contributes to hepatic damage during IR, which induces autophagy via extracellular signal-regulated kinease (ERK) activation [18]. However, another study demonstrates that necroptotic molecules are not increased in IR [19]. Apoptosis is another physiologic route to eliminate damaged or infected cells and to maintain tissue homeostasis, were caspase-3 is a central executor of apoptosis. Caspase-3 activity is highly important in the inhibition of cell proliferation. Caspase-3 plays an important role in suppressing proliferation in uncontrolled cell growth [20,21]. Canbek showed that B_2_O_3_ (an apoptosis inducer molecules) induced caspase-3 increase depending on time and inhibited NF-κB at the early stage of liver regeneration in a model of hepatectomy rat liver [20].

Macrophage phagocytosis of apoptotic cells and debris is a cellular hallmark of tissue resolution. Macrophages are heterogeneous cells that respond to tissue damage in a pleiotropic coordinated manner through either the classical inflammatory (M1) or alternative anti-inflammatory (M2) activation pathways. Two studies address whether pro-resolving molecules affect the macrophage infiltration via F4/80-positive cells. In the first, the authors evaluated the activity of MaR1 on adipocyte tissue inflammation, and found a decrease in the neutrophil infiltration, but not an enhancement of M2 macrophage polarization [22]. In the second study, the investigators showed, in a model of IR, that resolving (Rv) D1 promoted a reduction level of proinflammatory mediators, such as TNF-α and IL-6. The cytokine reduction occurred at 6 h of reperfusion while the neutrophil number started to reduce at 24 h after reperfusion. RvD1 increases hepatic F4/80 mRNA expression and the number of F4/80-positive cells, which may be linked to an increased population of restorative macrophages [23].

In our study, MaR1 modulated the levels of TNF-α and IL-6. Primary cytokines, such as TNF-α and IL-1, are generated by KCs during reperfusion. These cytokines recruit and activate CD4^+^ T-lymphocytes in the liver during the early reperfusion period. CD4^+^ T-lymphocytes produce mediators, such as TNF-β, IFN-γ, and granulocyte colony stimulating factor, which amplify the KC’s activation and promote neutrophil recruitment to the liver. TNF-α activates, via tumor necrosis factor receptor 1 (TNFRI) complex I, the modification/dissociation to a complex II, which include Fas-associated protein with death domain (FADD), and caspase-8/10 triggers apoptotic cell death [24,25]. Among proinflammatory cytokines, TNF-α exerts pleiotropic inflammatory and immunological functions by triggering the synthesis of downstream targets, such as IL-6 [26]. Previous studies have shown the ability of n-3 polyunsaturated fatty acids (PUFAs) to prevent inflammatory-cytokine effects [27,28], and recently, Laiglesia et al. described that MaR1 ameliorates TNF-α-induced alterations on lipolysis and autophagy in adipocytes. Moreover, MaR1 also decreased the ERK1/2 phosphorylation induced by TNF-α and reversed the TNF-α effects on specific autophagy-related proteins, including p62 and LC3I and II [14]. 

Inflammation is an important process of liver damage and boosts the regeneration of the injured liver. Cytokines such as TNF-*α* and IL-6, and growth factors released from inflammatory cells, triggered liver regeneration. These first events are recognized as a priming phase since they prime hepatocytes to re-enter into the cell cycle in the first stage and the loss of either TNF-*α* or IL-6 delay liver regeneration [29]. TNF-*α* acts as a pleiotropic molecule, because in the absence of Akt and NF-κB activation signaling, TNF-*α* mediates hepatocyte apoptosis and liver failure in mice [30]. A bulk of studies have highlighted the relation of MaR1 and the decrease of TNF-α, IL-6, and IL-1β [31,32,33,34], but they did not analyze the proliferatory potential of MaR1. Paradoxically, we found a decrease of TNF-α, but an increase in IL-6 at 3 h of perfusion, for that reason, we hypothesized that there was a prime event early in reperfusion (less than 1 h).

On the other side, IL-6 and its transmembrane protein receptor, Gp130, have been shown to have actions on cell proliferation, differentiation, and regulation of apoptosis, promoting anti-apoptotic effects [35]. It has been shown that IL-6 participates in the events of liver regeneration, promoting cell proliferation through bcl-2 [36]. Only 2 h after hepatectomy, the level of TNF-α increased, followed by a dramatic upregulation of IL-6 levels in the liver vein. After hepatectomy or liver damage, gut-derived factors like lipopolysaccharide (LPS) activate liver-resident KCs, resulting in a TNF-α-dependent secretion of IL-6 [35]. Also, the elevation of serum IL-6 levels during the first hours after hepatectomy generates a strong activation of the complex composed by NF-IL-6, a bZIP member of transcriptional factors associated to the induction of several cytokine genes and the enhancer binding protein C/EBPβ. The C/EBPβ/NF-IL-6 complex enhances transcription of genes and probably triggers G_0_/G_1_ phase transition of hepatocytes after hepatectomy [37,38,39]. Additionally, in an IL-6-deficient mouse model [40], liver failure occurred after hepatectomy. In these animals, the G_1_ phase was abnormal, which resulted in a reduced rate of DNA synthesis. From the aforementioned information, we can speculate that the inactivation of TNF-α and the observed increase in IL-6 would be orchestrated to allow the cell proliferation observed when MaR1 was administrated prior to the IR. We can hypothesize that the elevation of IL-6 observed in our study can be associated to an improvement of the early damage produce by the IR, but we cannot reject the theory of an hepatocellular carcinoma (HCC) phenotype related to enhanced cell proliferation because there is a study about the role of Il-6 in HCC [35]. Additionally, MaR1 administration enhanced IL-10, an important immunoregulatory cytokine produced by many liver cell populations, such as hepatocytes, sinusoidal endothelial cells, KCs, hepatic stellate cells, and liver-associated lymphocytes [41]. IL-10 protects against hepatic IR injury by suppressing NF-κB activation and subsequent expression of proinflammatory mediators, and it has been shown to be beneficial in the setting of liver disease and transplantation [42,43,44]. Recently, Yang et al., reported that MaR1 induced a sustained IL-10 response in a model of neuroinflammation, with fewer pro-inflammatory macrophages and a reduction of pro-inflammatory cell surface markers and cytokines by reduction of nuclear NF-κB signaling [45]. Similar results were found in a model of renal IR, where increases of IL-10 was related to an inhibition of the Toll-like receptor (TLR)4/MAPK/NF-κB [32], similar to that observed in a model of acute pancreatitis [46]. Interestingly, IL10 secreted by M2 KCs promoted selective M1 death in alcohol-exposed mice, where the neutralization of IL10 impaired M1 apoptosis [47]. With this in mind, it is possible to hypothesize that the anti-inflammatory role of MaR1 would be related to a polarization to a M2 KCs phenotype.

NF-κB has been shown to regulate several cellular functions, including inflammation and stress (oxidative or endoplasmic reticulum (ER))-induced responses [48], and ROS can directly activate NF-κB [49]. NF-κB activation is caused by oxidative stress, ER stress, or inflammation [48,49,50]. IR injury enhances the early (3 h) mobilization of NF-κB to the nucleus with the upregulation of TNF-α, IL-1β, and IL-6 expression in KCs [5,51,52]. Previously, it was found that omega-3 fatty acids’ administration can modulate the NF-κB DNA binding activity in a model of IR, which may allow for the recovery of the expression of NF-κB-controlling genes encoding for antioxidant, anti-apoptotic, and/or acute-phase response proteins [53]. There is scarce information about the role of MaR1 on this transcription factor. Ohuchi et al. investigated the effects of MaR1 on NF-κB activation in a model of several stress-induced motor neuron cell death. They observed that MaR1 has protective effects against several stresses by reduction of ROS production and attenuation of the NF-κB activation [19]. Interestingly, Xian et al. described a homeostatic and protective role of MaR1 on brain cerebral artery occlusion damage. MaR1 (1 ng per mice) reduced cerebral edema and the area of infarction through upregulation of Bcl-2, and a decrease in the levels of IL-6, IL-1β, and TNF-α, all related with a depletion of NF-κB nuclear translocation [33].

MaR1 administration can protect the liver against IR injury, a process related to severe ROS production. The molecular mechanism may involve the increase of Nrf-2 in the nucleus and be correlated with cell proliferation. Firat et al. reported the negative impact of oxidative stress via the production of ROS in cell proliferation, mainly due to the arrest of cell growth and activation of proteins that inhibit the cell cycle [54]. They found that the omega-3 parenteral administration to Wistar rats with right portal vein ligation, can reduce the levels of nitric oxide (NO) and ROS, with an increase in glutathione peroxidase and reduced glutathione (GSH), among other antioxidants’ systems of defense, improving the system of regeneration in the contralateral lobe [54]. Also, it is interesting to mention that NO protects hepatocytes from apoptosis after partial hepatectomy PH [55]. The transcription factor Nrf-2 is a key regulator of the antioxidant defense system, and pharmacological activation of Nrf-2 is an interesting strategy for prevention of toxin-induced liver damage and a promising candidate for liver protection under stress conditions [56]. Loss of Nrf-2 impairs liver regenerative capacity. Dayoub et al. reported that augmenter of liver regeneration (ALR), a hepatotropic molecule that supports the process of liver regeneration after partial hepatectomy, is regulated by Nrf-2, acts as a liver regeneration and antioxidative protein and, therefore, links oxidative stress to hepatic regeneration to ensure survival of damaged cells [57]. In this work, we found a possible relation between Nrf-2 enhancement mediated by MaR1 and the activation of liver cell proliferation. In a previous study, Sun et al. suggested the link between MaR1 and Nrf-2/OH-1 in lung IR injury through suppressing oxidative stress [34], reinforcing the idea that MaR1 could mediate these protective effects activating the Nrf-2 signaling and inhibiting the acute activation of NF-κB. In concordance with the above, Qiu et al. [32] determined that 1 ng of MaR1 injected thought the tail vein at the start of the reperfusion protects from renal IR and tissue dysfunction though a mitigation of inflammation and oxidative stress, by inhibiting the TLR4 and the expression of MAPK signaling pathways, decreasing NF-κB nuclear translocation, and the activation of nuclear NrF2.

In summary, our present study shows for the first time the protective action of MaR1 on IR liver injury. In the presence of MaR1, cell division is activated, and hepatocytes enter a mitotic activity, related to an increase of IL-6 and enhanced translocation of Nrf-2 to the nucleus with a decrease of NF-κB activity. This represents a potential therapeutic tool to treat liver IR damage.

## 4. Materials and Methods

### 4.1. Animals

Adult male Sprague-Dawley Rats were purchased from Bioterio Central Universidad de Chile, Santiago, Chile, with body weights ranging from 230 to 250 g. Animals were allowed free access to a regular rat chow (Champion S.A., Santiago, Chile), water ad libitum, and were housed on a 12 h light/dark cycle. All animal experiments were approved by the Institutional Animal Use Committee of the Universidad de Talca Nº2015-03-A from 12 November 2015, and conducted in accordance with the “Guide for the Care and Use of Laboratory Animals” (National Institute of Health (NIH) publication 86-23 revised 1985), and the recommendations of the European Union regarding animal experimentation (Directive of the European Council 86/609/EC). During the performance of the experimental protocol, all the animals were supervised for physiological functions by a veterinarian. 

### 4.2. Model of Partial Ischemia-Reperfusion (IR) Injury

For IR surgery, rats were anaesthetized with an intraperitoneal (i.p.) sedative-anesthetic mixture (ketamine 15 mg/kg, xylazine 3 mg/kg, and acepromazine 2.5 mg/kg), and IR was induced by temporally occluding the portal vein and hepatic artery branch that supply to the left and medial lobes of the liver with a microvascular Schwartz clamp for 1 h. Reperfusion was initiated by clamp removal and animals were sutured to avoid evisceration, animals were sacrificed at 3 h after reperfusion as described [58]. During surgery, the temperature of animals was kept at 32–33 °C using a warming surgical platform and the viscera was hydrated with a saline solution. The animals did not receive any extra dose of anesthesia during reperfusion. Control animals were subjected to anesthesia and sham laparotomy. Animals were divided into four groups: (1) the sham group (control), who received isovolumetric vehicle: NaCl 0.9%, and then were subjected to a sham laparotomy, (2) the IR group, where animals received the same vehicle as the sham group and then were submitted to IR surgery, (3) the MaR1-sham group, where rats received i.p. a single dose of 4 ng/g per body weight of MaR1 (Cayman Chemical, Ann Arbor, MI, USA) and sham surgery, and (4) the MaR1-IR group, where one-hour prior to the IR, animals were injected with MaR1 or vehicle. To evaluate the action of MaR1, blood samples were obtained by cardiac puncture and liver was extracted and cut in to two pieces after reperfusion. One half was fixed by 10% formalin and used for histopathological analysis. The other halves were stored at −80 °C for biochemical analyses. MaR1 doses used were selected according to the mean values and the results reported by Marcon et al., and Li et al. [7,8].

### 4.3. Measurement of Serum Parameters

Serum was obtained by blood clotting for 3 h at room temperature and centrifugated at 2000× *g* for 5 min. Serum aspartate aminotransaminase (AST) and alanine transaminase (ALT) activities were measured using specific diagnostic kits (Valtek Diagnostics, Santiago, Chile). Enzyme-linked immunoSorbent assay (ELISA) kits were used for the assessment of TNF-α, IL-6, and IL-10 (Thermo Scientific, Waltham, MA, USA), following the manufacturer’s instructions.

### 4.4. Histopathological Examination

Halves of liver were fixed in 10% phosphate buffered formalin for 24 h, embedded in paraffin, and cut in 5 μm thickness sections and stained with hematoxylin-eosin (H and E) for morphology assessment. The examination was performed by a pathologist (IC) in a blind manner. Five random fields were assessed for (i) necrosis, which was evaluated according to the Korourian score [59]: none = 1, occasional (1%) necrotic hepatocyte = 1, frequent (5–10%) necrotic hepatocytes = 2, small foci of necrosis (clusters greater than 10 necrotic hepatocytes) = 3, and extensive areas of necrosis (over 25%) = 4. (ii) Focal and portal inflammation, according to Goodman’s adapted Ishack score [60,61]: none = 0, one focus per 10× objective or less and/or mild inflammation in portal area = 1, two to four foci per 10× objective and/or moderate, some or all portal areas = 2, five to ten foci per 10× objective and/or moderate/marked inflammation in all portal areas = 3, and more than 10 foci per 10× objective and/or marked inflammation in all portal areas = 4. (iii) Architecture criteria was analyzed according Goodman’s [60] schematic diagram: no changes = 1, mild interphase hepatitis and parenquimal injury but central vein conserved = 2, moderate interface hepatitis with necroinflammatory foci and moderate parenquimal injury and more of the 50% of central vein integrity = 3, marked interface hepatitis, stromal collapse with indicators of parenquimal injury and loss of more than 50% of central vein integrity = 4. (iv) Mitotic activity index (MAI) was analyzed according to Al-Janabi [62]: twenty areas per tissue slide (fields) were marked (avoiding inclusion of apoptotic and necrotic cells) and all results are shown as the average of the results obtained in these fields. The counting cells were observed at 400× magnification. The total hepatocyte count per fields and mitosis cell was obtained. Then, MAI was reported as a percentage of total hepatocytes. All the histological analyses were performed in a Nikon Eclipse 50i Optic microscope (Nikon, Tokyo, Japan), the posterior analysis and photograph were developed in Micrometrics SE Premium™ software (Opticstar, Manchester, UK).

### 4.5. Immunofluorescense Staining

For immunofluorescence, 5 μm thickness of liver sections were put in xilanizated slides and boiled in sodium citrate buffer pH 6 (10 mM sodium citrate, 0.05% Tween 20), quenched with 100 mM glycine in phosphate buffer saline (PBS), and incubated with: (i) mouse F4/80 monoclonal antibodie 1:20 (Santa Cruz, USA), followed by fluorescein isothiocyanate conjugated (FITC) anti-mouse (Jackson Inmunoresearch, West Grove, PA, USA), and counterstained with 100 µM propidium iodine. (ii) Rabbit polyclonal Ki67 antibodie 1:300 (Merck Millipore), followed with tetramethylrhodamine conjugated (TRITC) anti-rabbit (Jackson Inmunoresearch, West Grove, PA, USA) and counterstained with Sytox^TM^ 11 green fluorescent 1:10,000 (Thermo Scientific, Waltham, MA, USA). Analyses were performed in a confocal microscope LSM700 (Carl Zeiss, Jena, Germany). Fluorescence was determined in Zen software (Carl Zeiss, Jena, Germany). A minimum of 6 fields per slides were analyzed.

### 4.6. Western Blot Analysis

Cytoplasmic and nuclear samples were obtained from the frozen hepatic tissue samples (100 mg), and the assay was realized from the adapted protocol of Deryckere and Gannon [63]. 200–500 mg of frozen liver were homogenized and suspended in buffer solution pH 7.9 (10 mM 4-(2-hydroxyethyl)-1-piperazineethanesulfonic acid (HEPES), 1 mM ethylenediaminetetraacetic (EDTA), 0.6% nonidet P (NP)-40, 150 mM NaCl, and 0.5 mM phenyl methyl sulphonyl fluoride (PMSF)), followed by centrifugation at 480× *g* for 15 s at 4 °C, the supernatant was incubated in ice for 5 min and centrifugated at 3020× *g* for 5 min at 4 °C. The supernatant corresponds to cytoplasmic fractions, and the precipitate was resuspended in 200 µL of nuclear buffer solution pH 7.9 (20 mM HEPES, 0.2 mM EDTA, 25% glycerol, 420 mM NaCl, 1.2 mM MgCl_2_, 0.5 mM di-thio threitol DTT, 0.5 mM PMSF, and 2 mM benzamidine, a protease inhibitors cocktail (Pierce, Thermo Scientific)), followed by centrifugation at 13,000× *g* for 60 s (the supernatant correspond to nuclear proteins), and incubated for 20 min in ice, and then was centrifugated at 13,000× *g* for 30 s at 4 °C. Cytoplasmic and nuclear protein fractions (50 μg) were separated on 12% polyacrylamide gels using sodium dodecyl sulfate polyacrylamide gel electrophoresis (SDS-PAGE) and transferred to nitrocellulose membranes, which were blocked for 3 h at room temperature with tris buffer saline (TBS)-containing skim milk. Rabbit polyclonal primary antibodies used were: anti-Nrf-2 (1:500), anti-NF-κB (1:500), and anti-histone H1 (1:250) as a nuclear housekeeping protein. Mouse monoclonal primary antibodies used were: anti-RIP3 (1:200), anti-cleaved caspase-3 (1:1000), and anti-glyceraldehyde 3-phosphate dehydrogenase (GAPDH) (1:2000) was used as a cytoplasmic housekeeping protein. The antibodies were incubated overnight at 4 °C. After extensive washing, the antigen antibody complexes were detected using horseradish peroxidase-labeled goat anti-rabbit IgG/anti-mouse or rabbit (all antibodies used in Western blot were purchased from Merck Millipore, USA) and the protein was detected with the kit of protein detection Amersham enhanced chemioluminicense (ECL) Prime Western Blotting Detection Reagent (General Electric Healthcare, Hammersmith, UK). The chemiluminescent signals were analyzed in the Omega Lum^TM^ System (Aplegen, San Francisco, CA, USA), and the quantification of luminescent images was made in ImageJ (NIH, Bethesda, MD, USA).

### 4.7. Statistical Analysis

Values shown correspond to the mean ± standard deviation (SD) or SEM for each experimental group. One-way analysis of variance (ANOVA) with the Tukey’s test as a post-hoc test was used to assess differences between means of the different groups. For non-parametric data, the Kruskal–Wallis or Mann–Whitney test were used. A *p*-value of less than 0.05 was considered significant. The analyses were performed using the GraphPad Prism 6.0 software (GraphPad software, Inc. San Diego, CA, USA).

## Figures and Tables

**Figure 1 ijms-21-00540-f001:**
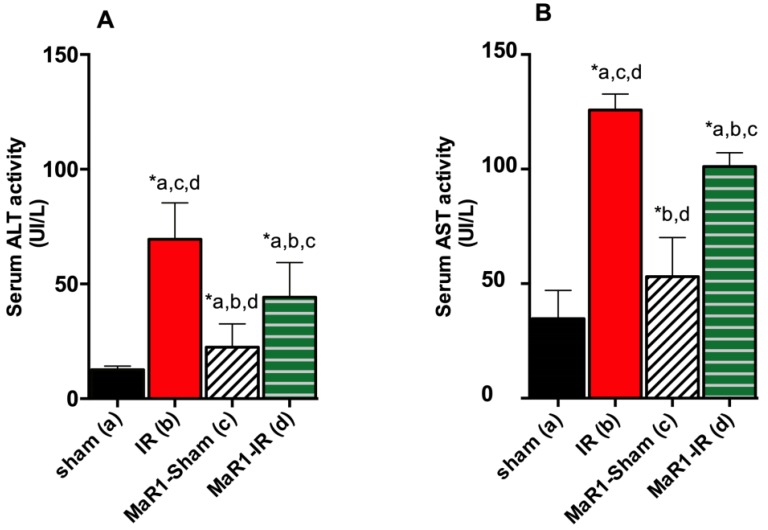
Effect of maresin-1 (MaR1) on serum (**A**) alanine aminotransferase (ALT) and (**B**) aspartate aminotransferase (AST) levels after liver ischemia (1 h)-reperfusion (3 h) (IR). Values correspond to means ± SEM of 9 animals per experimental group. Significance was assessed by one-way analysis of variance (ANOVA) and the Tukey’s post-test. Asterisk indicates *p* < 0.05, and the letters identify the experiments that are compared and present this statistical difference.

**Figure 2 ijms-21-00540-f002:**
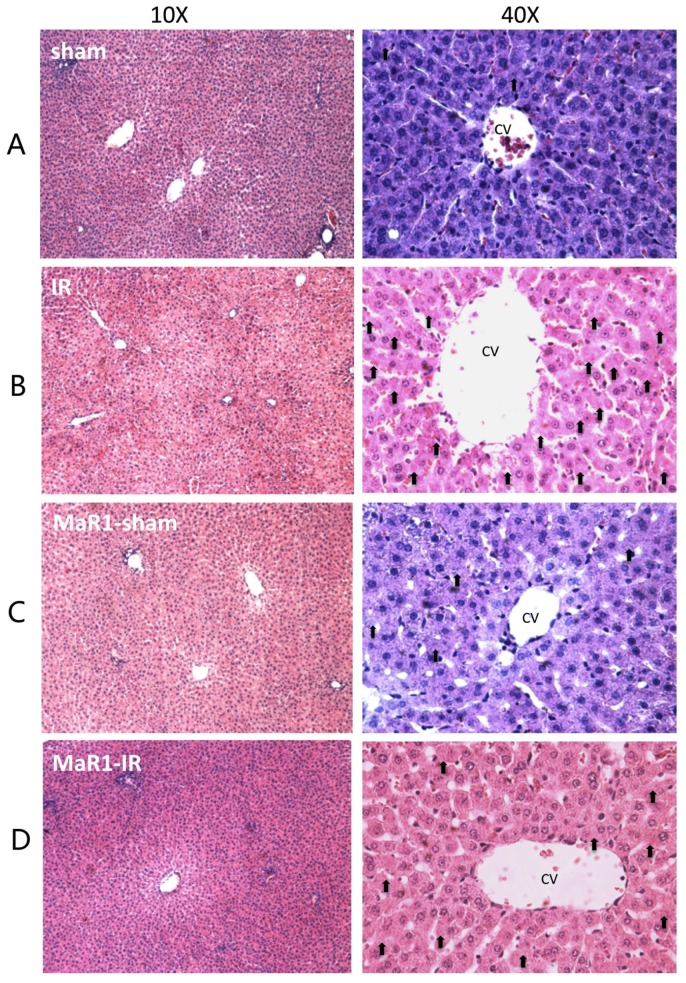
Effect of MaR1 on liver morphology after IR. (**A**–**D**) Representative liver sections stained with hematoxylin-eosin (H and E), magnification 100× and 400×, scale bar = 100 μm, from a total of 9 animals per experimental group. Scores of liver H and E sections were graphed for (**E**) architecture, (**F**) inflammation and (**G**) necrosis. Values correspond to the means ± standard deviation (SD) of 9 animals per experimental group. Asterisk indicates *p* < 0.05 and the letters identify the experiments that are compared and present this statistical difference. CV: Central vein. Black arrow indicates focal necrosis hepatocytes.

**Figure 3 ijms-21-00540-f003:**
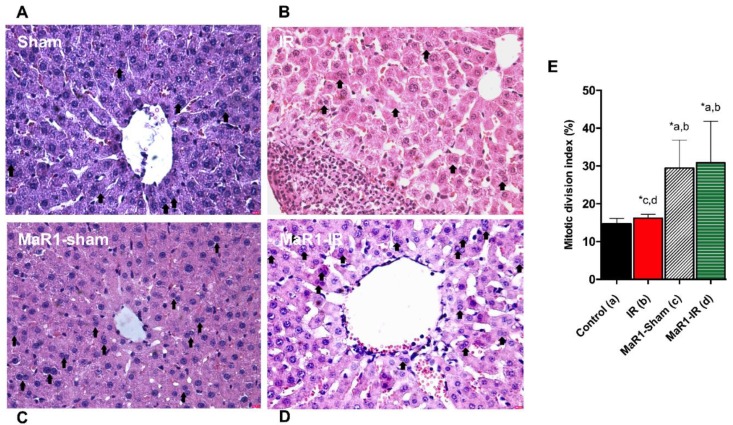
Impact of MaR1 on cell proliferation after IR. Representative microphotography of (**A**) Sham, (**B**) IR, (**C**) MaR1-sham, (**D**) MaR1-IR, and (**E**) mitotic activity index (MAI) measured as a percentage of a whole section, 20 fields for every sample were analyzed at 400× magnification. *n* = 9 animals per experimental group. Asterisk indicates *p* < 0.05, and the letters identify the experiments that are compared and present this statistical difference.

**Figure 4 ijms-21-00540-f004:**
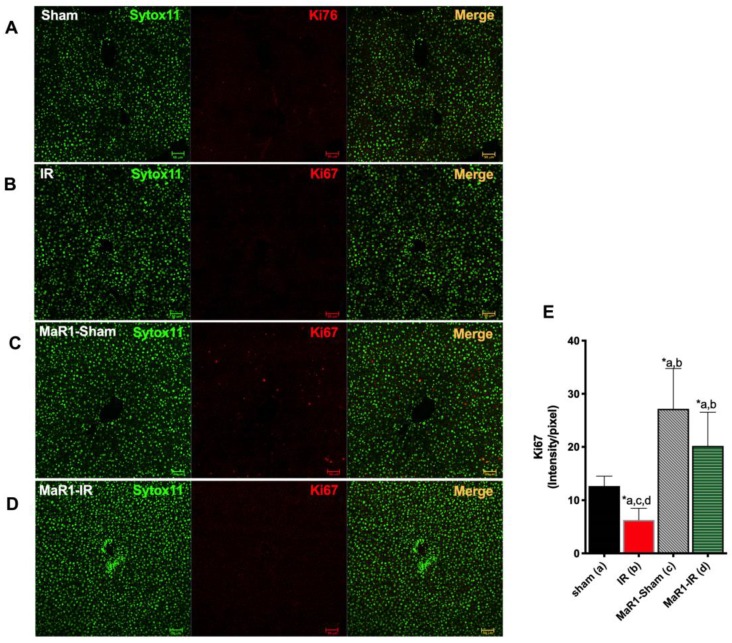
Effect of MaR1 on cell proliferation measure thought Ki67 immunofluorescence red signal and nuclei in green (Sytox^TM^ Green 11). Representative images of (**A**) sham, (**B**) IR, (**C**) MaR1-IR, and (**D**) MaR1-IR. (**E**) The Ki67 signal intensity was evaluated. The plot is represented as mean ± SEM, *n* = 4 rats per experiment, Image fields measured per rats >6. Significance was assessed by one-way ANOVA and the Tukey’s post-test. Asterisk indicates *p* < 0.05, and the letters identify the experiments that are compared and present this statistical difference. Scale bar indicates 50 µm.

**Figure 5 ijms-21-00540-f005:**
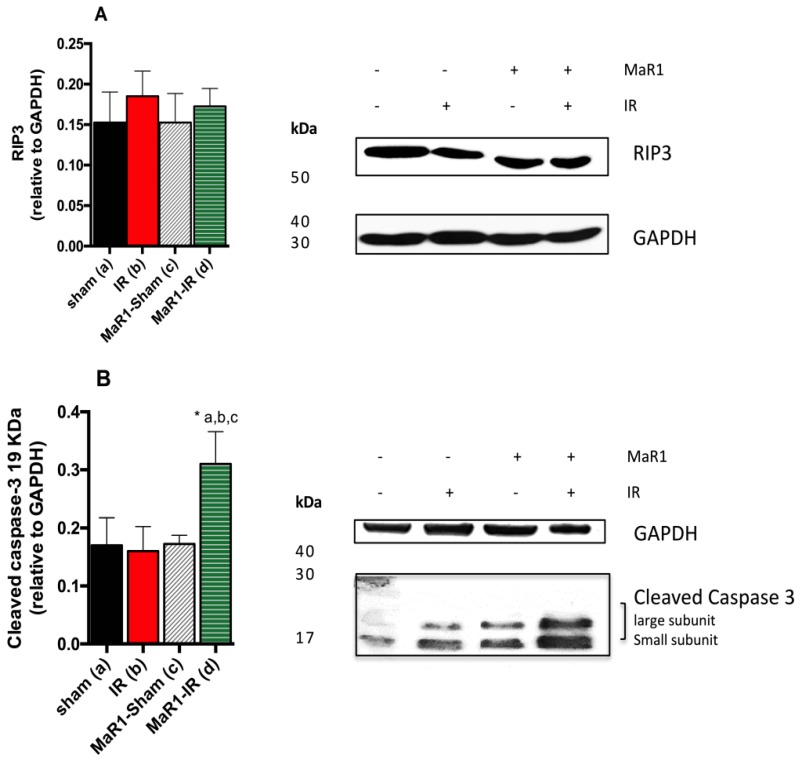
Effect of MaR1 after IR on tissue levels of (**A**) RIP3 and (**B**) cleaved caspase-3 protein expression levels. The values were quantified relative to glyceraldehyde 3-phosphate deshydrogenase (GAPDH). *n* = 4 rats per experimental group. Asterisk indicates *p* < 0.05, and the letters identify the experiments that are compared and present this statistical difference.

**Figure 6 ijms-21-00540-f006:**
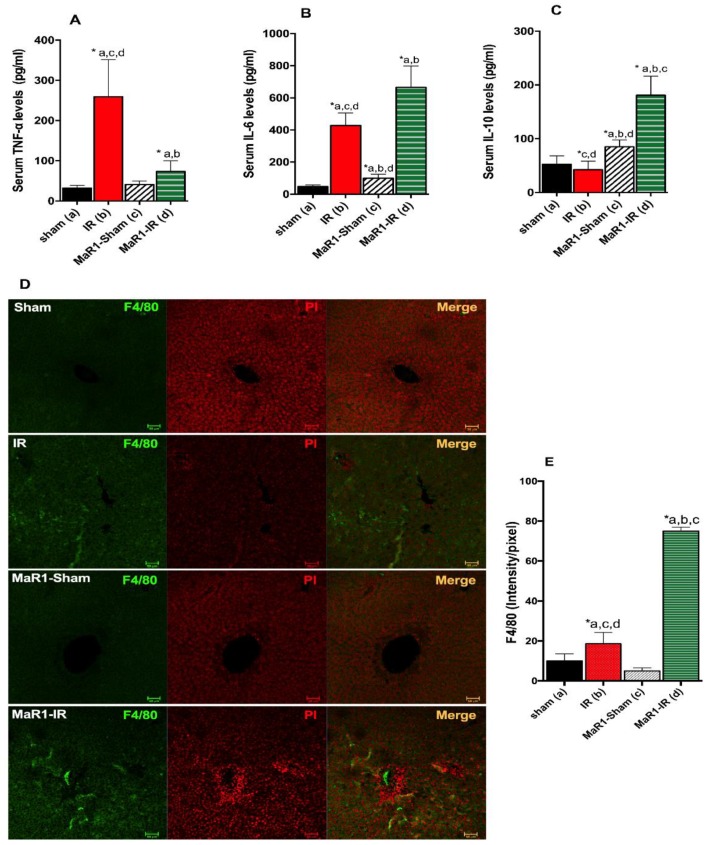
Effect of MaR1 on inflammatory response. Serum levels of inflammatory cytokines (**A**) tumor necrosis factor (TNF)-α, (**B**) interleukin (IL)-6 and anti-inflammatory cytokine, and (**C**) IL-10 levels after IR were quantified. (**D**) Representative images depicting hepatic macrophages stained with the F4/80 marker in green and nuclei in red (propidium iodine), assessed by immunofluorescence in paraffined fixed liver tissues. (**E**) The fluorescence intensity was evaluated. The plots are represented as mean ± SEM, *n* = 4 rats per experiment, Image fields measured per rats >6. Significance was assessed by one-way ANOVA and the Tukey’s post-test. Asterisk indicates *p* < 0.05, and the letters identify the experiments that are compared and present this statistical difference. Scale bar indicates 50 µm.

**Figure 7 ijms-21-00540-f007:**
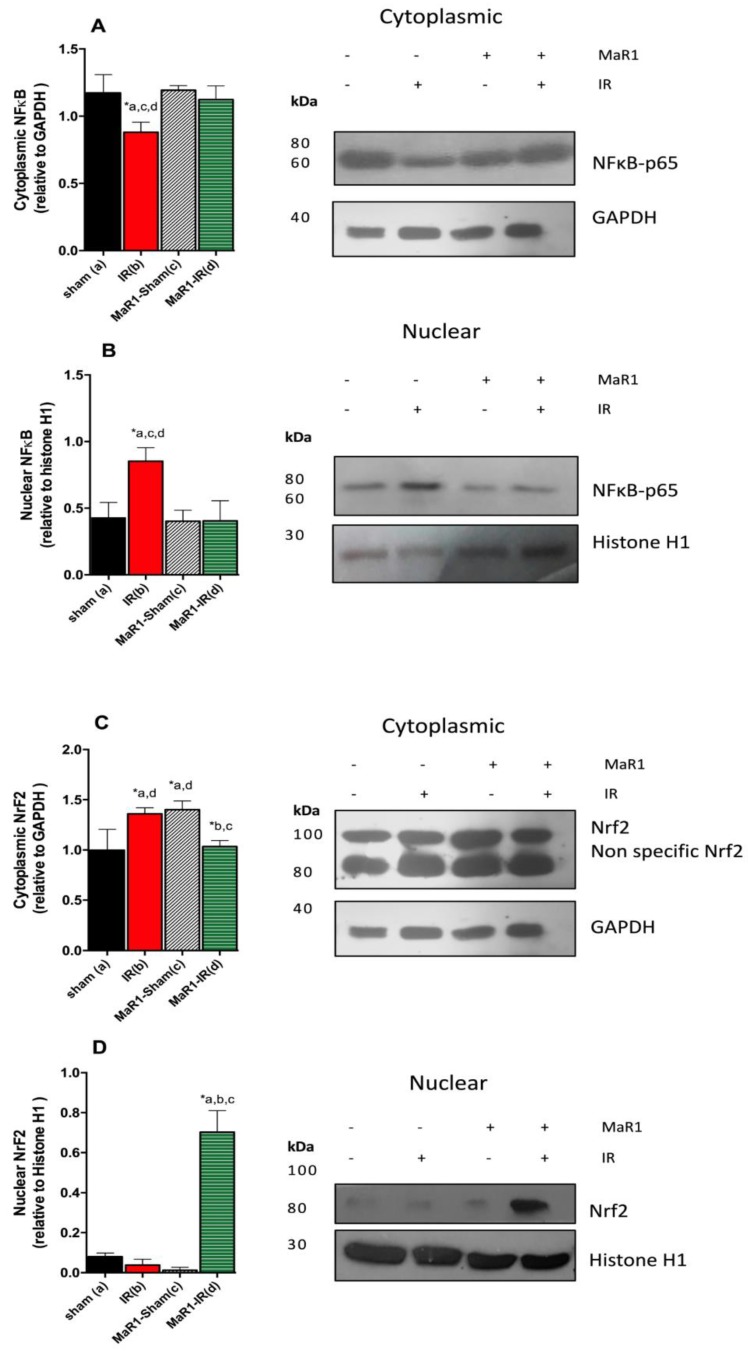
Effect of MaR1 on tissue levels of Nrf-2 and NF-κB after IR. (**A**,**B**) represents results cytoplasmic and nuclear levels of NF-κB transcription factor. (**C**,**D**) represents cytoplasmic and nuclear levels of Nrf-2 transcription factor. The cytoplasmic levels were quantified relative to GAPDH as housekeeping and nuclear levels were quantified relative to histone H1 as housekeeping. *n* = 9 rats per experimental group. Asterisk indicates *p* < 0.05, and the letters identify the experiments that are compared and present this statistical difference.

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
