# Peer review of "Maresin 1, a Proresolving Lipid Mediator, Ameliorates Liver Ischemia-Reperfusion Injury and Stimulates Hepatocyte Proliferation in Sprague-Dawley Rats"

_ijms, 2020, doi:10.3390/ijms21020540_

Round 1
Reviewer 1 Report
Great improvement from the initial submission.
Author Response
Thanks for you valuable comments and review.
Reviewer 2 Report
An explanation of Maresin 1 dose used in this study should be reported in the method section.
The histograms of figure 5 and 7 are too small; enlarge them.
Minor spell check required: line 170:Mar-1, line 174 MaR1 and line 195: Mar!. Check the manuscript
Author Response
Dear Reviewer,
Thanks for your valuable comments and review of the manuscript. We modify the text as requested:
MaR1 dose was informed in methodology --> LINE 101-102
Histograms are sent in tiff format, to make sure they have the appropriate size.
Minor spelling was checked, the error on line 170 and 174 was corrected.
This manuscript is a resubmission of an earlier submission. The following is a list of the peer review reports and author responses from that submission.
Round 1
Reviewer 1 Report
The authors reported the effects of Maresin-1 in a rat model of hepatic I/R injury: a reduction in liver damage and an increased hepatocyte proliferation were detected.
Abstract: lines 22-23 and lines 25-28 not clear, should be rewritten.
Introduction: Lines 37-39: The authors should explain that the postoperative graft dysfunction after liver transplantation is associated with cold ischemia time, during graft preservation, and warm ischemia time, defined as a rewarming time.
Methods: Group 1 should be defined as “Sham” instead of “Control”.
Page 2, line 89: explain the identification of MarR1 dose.
The isolation of cytoplasmic fraction and the nuclear protein should be better explained and a reference should be added.
The histopathological examination should be better described: the authors should report how they have measured necrosis, inflammation and architecture criteria as well as mitotic activity index. The microscope equipment should be described.
Results: Table 1 should be eliminated and data reported in the result section.
In figure 1 a significant increase in ALT was found in MaR1-Sham: this result should be reported and explained.
Figure 2: New figures with increased magnification should be added to explain the results reported in panel E, F and G. Add arrows in the figures.
Figure 3: the magnification used should be reported.
“The MaR1-IR group normalizes the morphological damage induced by IR in a 32%, 42% and 50% for the architecture, inflammation and necrosis respectively” not clear, describe the results without the use of the percentage .
“Levels of the anti-inflammatory IL-10 (Figure 4C) showed no differences among groups”: not true, describe the results.
Page 7, lines 187-188: ….IR group showed basal level of Nrf-2: not clear (figure 5E, IR *a); eliminate “(value near to zero)”
The sentence, page 7, lines 198-199, should be eliminated, only reported in the legends.
Discussion Page 8: Line 220, explain “certain circumstances” and add a reference. Line 231, “C/EBPβ/NF-IL-6” not clear.
The discussion should be improved. The authors should compare the results of MaR1 effects in liver I/R (i.e. on TNF-alpha, IL-6, IL-10 and NF-kB content) with those recently published on MaR-1 effects in renal and cerebral I/R injury. Qiu Y et al. (2019) and Xian W et al. (2019) have reported a decrease in tissue IL-6 levels after MaR1 treatment.
References
Qiu Y et al. Maresin 1 mitigates renal ischemia/reperfusion injury in mice via inhibition of the TLR4/MAPK/NF-κB pathways and activation of the Nrf2 pathway. Drug Des Devel Ther. 2019;13:739-745. doi: 10.2147/DDDT.S188654 Xian W et al. Maresin 1 attenuates the inflammatory response and mitochondrial damage in mice with cerebral ischemia/reperfusion in a SIRT1-dependent manner. Brain Res. 2019;1711:83-90. doi: 10.1016/j.brainres.2019.01.013
Reviewer 2 Report
The topic is very interesting with potential interest in healthcare and medicine. Even though I have some concerns regarding the results of this study.
MAJOR CONCERNS
MaR1: As you mentioned, this compound was recently discovered. For this reason it is necessary to be more specific about the formula and its providers. Is there any report about its pharmacokinetic?. Is this compound (the formula you used) completely absorbed after one hour of its i.p. administration? These are crucial questions about the methodology which should be addressed.
Histological evidences:
-The features analysed for the liver architecture histological assessment are not properly detailed in material and methods section. You do not mention if the analysis was performed by one o more investigators or pathologists in a blinded manner. The “Korouvax” method that you referenced clearly specify that the observation was performed by a pathologist. Korouvax do not describe the features for considering i.e. “ballooning” which is a common feature in ischemic hepatocytes. We consider also unclear the reported results regarding mitosis for the same reason. A magnification of the field is required in both cases.
I recommend the authors to re-evaluate the liver with a pathologist assistance. I also recommend them to read the following article for more information about liver histological score criteria:
Journal of Hepatology 47 (2007) 598–607
MINOR CONCERNS
You have omitted information about the surgical procedure. The surgery ended at least 4h after laparotomy, Did you add more anaesthesia between this procedure?.
Did you maintain body temperature and guaranteed hydration of visceral organs? Was the liver perfused before extracting?
When you say “serum” it is obtained after blood clotting, is that correct?. If you pre-treated blood samples for avoid coagulation you should refer the supernatant as “plasma”.
Reviewer 3 Report
Methods:
-Unclear when blood samples were taken, was this at endpoint 3 hours post reperfusion?
-Was the abdomen closed immediately following reperfusion or was the animal left open?
Results:
-Table 1 is unnecessary.
-Figure 1 results show a modest decrease in AST and ALT between IR and MaR1-IR groups however, it appears that the error bars overlap. Might be helpful to include numeric values.
-In Figure 2, authors quantify inflammation and necrotic foci but do not show any to the read. Also, reviewer disagrees with the way both inflammation and necrosis were determined. A blinded pathologist should determine this. Would be helpful to include H&E pictures of inflammation and necrotic foci like those shown in Figure 3 for mitotic activity.
-Authors should use IF and stain for leukocytes and Kupffer cells to provide additional evidence for inflammation.
-Line 160 state the Korourian scale was adapted, how so from the original 1999 paper?
-The liver is an organ that can regenerate, given previous results from Figure 1 which stated IR group had the most damage (highest ALT/AST), how does this group have similar proliferation as the control? Enormous amount of literature shows hepatic IR injury causes proliferation, it would be expected the IR group would have more proliferation. Ki67 staining may be a better way to measure proliferation.
-Survival experiments should be included in this study. If the animals all may a full recover after IR injury, what is the purpose of adding MaR1?
Line 169 - states that inflammatory cytokines were evaluated, however in Figure 4, IL-10 an anti-inflammatory cytokine is included, should restate and correct Figure 4 title.
-Was there a significant increase in TNF-a in MaR1-IR compared to group?
-Figure 4B should read *a, b,d not c above MaR1-Sham.
-Line 173 - authors state there was no difference in the levels of IL-10 among groups hwoever, there is a clear increase in serum IL-10 in both MaR1-Sham and MaR1-IR. This point grossly overlooked.
-Authors should repeat Western blots nuclear fractions blotting for NrF-2 and loading control. The loading control is barely visible.
-Unclear how authors quantified Western blots, nothing is stated in the methods.
-Replace "y" with 'and' in line 142.
-Line 180 shows NrF-2 then in lines 187-189 and in Figure 5, stated as Nrf-2.
-Lines 198 and 199 should be in Figure 5 legend.
-Authors state MaR1 is the reason for the decrease in liver damage generated by IR related to pro-inflammatory cytokine production, however results presented in this paper show MaR1 increase the pro-inflammatory cytokine IL-6 along with decreases in nuclear NFkB-p65.
-Line 212 is a run on sentence
-Line 217 - Hepatocytes respond to TNFa produced by KCs during IR injury will either proliferate or undergo apoptosis/necroptosis. Would the levels of TNFa decreased by MaR1 decrease survival?
-TNFa produced by KC post-IR injury are known to induce proliferation in hepatocytes through TNFR1 in the NFkB pathway. However, in this manuscript results, TNFa is decreased with a corresponding decrease in nuclear NFkB-p65 which may lead to decrease survival. Authors should at least blot for caspase-3 to determine whether cells are undergoing apoptosis and could blot for RIP3 for necroptosis.
-Line 237 - Unclear why comments on hepatocellular carcinoma are found in the discussion.
-Li et al 2016 [Maresin 1 mitigates inflammatory response and protects mice from sepsis] investigated MaR1 administration in a sepsis mouse model showed decreased serum TNFa and IL6 while inhibiting NF-kB pathway in peripheral blood. Perhaps a more thorough literature search would provide authors would a more comprehensive background on current studies involving MaR1.
-Authors never discuss the increase in serum IL-10.